# Evaluating and Calibrating Uncertainty Prediction in Regression Tasks

## Abstract

Predicting not only the target but also an accurate measure of uncertainty is important for many applications and in particular safety-critical ones. In this work we study the calibration of uncertainty prediction for regression tasks which often arise in real-world systems. We show that the existing definition for calibration of a regression uncertainty (Kuleshov et al., 2018) has severe limitations in distinguishing informative from non-informative uncertainty predictions. We propose a new definition that escapes this caveat and an evaluation method using a simple histogram-based approach inspired by reliability diagrams used in classification tasks. Our method clusters examples with similar uncertainty prediction and compares the prediction with the empirical uncertainty on these examples. We also propose a simple scaling-based calibration that preforms well in our experimental tests. We show results on both a synthetic, controlled problem and on the object detection bounding-box regression task using the COCO (Lin et al., 2014) and KITTI (Geiger et al., 2012) datasets.

## 1 Introduction

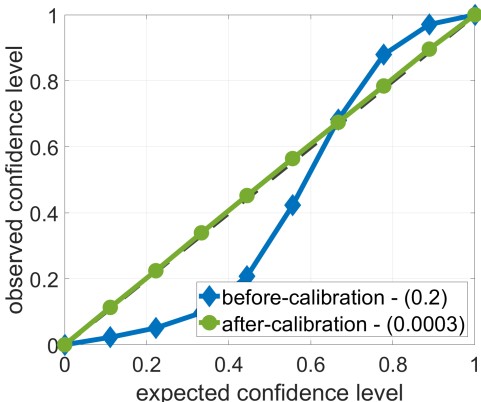

Figure 1: Regression with **random uncertainty** (independent of actual uncertainty) almost perfectly calibrated by the method proposed in (Kuleshov et al., 2018), when the expected and observed confidence level are identical. As anything can be perfectly calibrated, this calibration definition becomes uninformative. The task is object bounding box regression, using the KITTI dataset (Geiger et al., 2012). See details in Section 4.2.

Regression problems arise in many real-world machine learning tasks. To name just a few: Depth from a single image (Eigen et al., 2014), Object localization and Acoustic localization (Vera-Diaz et al., 2018). Many of these tasks are solved by deep neural networks used within decision making pipelines which require the machine learning block not only to predict the target but to also output its confidence in the prediction. For example, the commonly used Kalman-Filter tracking algorithm (Blackman, 2004) requiring variance estimation for the observed object's location estimation. In addition, we may want the system to output a final uncertainty, reflecting real-world empirical

probabilities, to allow a safety-critical system such as a self-driving car agent to take appropriate actions when confidence drops. In practice, using the confidence in the localization of objects has been shown to improve the non-maximal suppression stage and consequently the overall detection performance (He et al., 2018). Similarly, (Feng et al., 2018) describe a probabilistic 3D vehicle detector for Lidar point clouds that can model both classification and spatial uncertainty.

To provide uncertainty estimation, each prediction produced by the machine learning module during inference should be a distribution over the target domain. There are several approaches for achieving this, most common are Bayesian neural networks (Gal, 2016; Gal & Ghahramani, 2016), ensembles (Lakshminarayanan et al., 2017) and outputting a parametric distribution *directly* (Nix & Weigend, 1994). Bayesian neural networks place a probability distribution over the network parameters, which is translated to an uncertainty in the prediction, providing a technically sound approach but with overhead at inference time. In the *direct* approach, outputs of the network represent the parameters of the output distribution for either discrete (Niculescu-Mizil & Caruana, 2005) or continuous (Nix & Weigend, 1994) distributions. Note that the direct approach naturally captures the aleatoric uncertainty (inherent observation noise), but captures less the epistemic uncertainty (uncertainty in the model) (Kendall & Gal, 2017). We chose as a test case for our calibration method, the *direct* approach for producing uncertainty: we transform the network output from a single scalar to a Gaussian distribution by taking the scalar as the mean and adding a branch that predicts the standard deviation (STD) as in (Lakshminarayanan et al., 2017). While this is probably the simplest form, it is commonly used in practice, and our analysis is applicable to more complex distributions as well as other approaches.

Adjusting the output distributions to match the observed empirical ones via a post process is called *uncertainty calibration*. It was shown that modern deep networks tend to be over confident in their predictions (Guo et al., 2017). The same study revealed that for classification, Platt Scaling (Platt, 1999), a simple scaling of the pre-activation of the last layer, achieves well calibrated confidence estimates (Guo et al., 2017). In this paper we show that a similar simple scaling strategy, applied to the standard deviations of the output distributions, can calibrate regression algorithms as well.

One major question is how to define calibration for regression, where the model outputs a continuous distribution over possible predictions. In recent work (Kuleshov et al., 2018) suggested a definition based on credible intervals where if we take the $p$ percentiles of each predicted distribution the output should fall below them for exactly $p$ percent of the data. Based on this definition the authors further suggested a calibration evaluation metric and re-calibration method. While this seems very sensible and has the advantage of considering the entire distribution, we found serious flaws in this definition. The main problem arises from averaging over the whole dataset. We show, both empirically and analytically, that one can calibrate using this evaluation metric practically any output distribution, even one which is entirely uncorrelated with the empirical uncertainty as can be seen in Fig. 1. We elaborate on this property of the evaluation method described in (Kuleshov et al., 2018) in Section 2 and show empirical evidence in Section 4.

We propose a new simple definition for calibration for regression, which is closer to the standard one for classification. Calibration for classification can be viewed as expecting the output for every *single data point* to correctly predict its error, in terms of misclassification probability. In a similar fashion, we define calibration for regression by simply replacing the misclassification probability with the mean square error. Based on this definition, we propose a new calibration evaluation metric similar to the Expected Calibration Error (ECE) (Naeini et al., 2015), which groups examples into interval bins with similar uncertainty, and then measures the discrepancy between each bin's parameters and the parameters of the empirical distribution within the bin. An additional dispersion measure completes our set of diagnostic tools by revealing cases where the individual uncertainty outputs are uninformative as they all return similar values.

Finally, we propose a calibration method where we re-adjust the predicted uncertainty, in our case the outputted Gaussian variance, by minimizing the negative-log-likelihood (NLL) on a separate re-calibration set. We show good calibration results on a real-world dataset using a simple parametric model which scales the uncertainty by a constant factor. As opposed to (Kuleshov et al., 2018), we show that our approach cannot calibrate predicted uncertainty that is uncorrelated with the real uncertainty, as one would expect.

To summarize, our main contributions are:

- Revealing the fundamental flaws in the current definition of calibrated regression uncertainty (Kuleshov et al., 2018)

- A new proposed definition for calibrated uncertainty in regression tasks

- A simple scaling method that can reduce the calibration error similar to temperature scaling for classification (Guo et al., 2017), evaluated on large scale real world vision datasets.

## 1.1 RELATED WORK

While shallow neural networks are typically well-calibrated (Niculescu-Mizil & Caruana, 2005), modern, deep networks, albeit superior in accuracy, are no-longer calibrated (Guo et al., 2017). Uncertainty calibration for classification is a relatively studied field. *Calibration plots* or *Reliability diagrams* provide a visual representation of uncertainty prediction calibration (DeGroot & Fienberg, 1983; Niculescu-Mizil & Caruana, 2005) by plotting expected sample accuracy as a function of confidence. Confidence values are grouped into interval bins to allow computing the sample accuracy. A perfect model corresponds to the plot of the identity function. The *Expected Calibration Error (ECE)* (Naeini et al., 2015) summarizes the reliability diagram by averaging the error (gap between confidence and accuracy) in each bin, producing a single value measure of the calibration. Similarly, the *Maximum Calibration Error (MCE)* (Naeini et al., 2015) measures the maximal gap. *Negative Log Likelihood (NLL)* is a standard measure of a model's fit to the data (Hastie et al., 2001) but combines both accuracy of the model and its uncertainty estimation in one measure. Based on these measures, several calibration methods were proposed, which transform the network's confidence output to one that will produce a calibrated prediction. Non-parametric transformations include Histogram Binning (Zadrozny & Elkan, 2001), Bayesian Binning into Quantiles (Naeini et al., 2015) and Isotonic Regression (Zadrozny & Elkan, 2001) while parametric transformations include versions of Platt Scaling (Platt, 1999) such as Matrix Scaling and Temperature Scaling (Guo et al., 2017). In (Guo et al., 2017) it is demonstrated that the simple Temperature Scaling, consisting of a one scaling-parameter model which multiplies the last layer logits, suffices to produce excellent calibration on many classification data-sets.

In comparison with classification, calibration of uncertainty prediction in regression, has received little attention so far. As already described, (Kuleshov et al., 2018) propose a practical method for evaluation and calibration based on confidence intervals and isotonic regression. The proposed method is applied in the context of Bayesian neural networks. In recent work (Phan et al., 2018), the authors follow (Kuleshov et al., 2018) definition and method of calibration for regression, but use a standard deviation vs. MSE scatter plot, somewhat similar to our approach, as a sanity check.

## 2 CONFIDENCE-INTERVALS BASED CALIBRATION

We next review the method for regression uncertainty calibration proposed in (Kuleshov et al., 2018) which is based on confidence intervals, and highlight its shortcomings. We refer to this method in short as the "interval-based" calibration method. We start by introducing basic notations for uncertainty calibration used throughout the paper.

**Notations**. Let $X, Y \sim \mathrm{P}$ be two random variables jointly distributed according to P and $\mathcal{X} \times \mathcal{Y}$ their corresponding domains. A dataset $\{(x_t, y_t)\}_{t=1}^T$ consists of i.i.d samples of $X, Y$. A forecaster $H : \mathcal{X} \to \mathcal{P}(\mathcal{Y})$ outputs per example $x_t$ a distribution $p_t \equiv H(x_t)$ over the label space, where $\mathcal{P}(Y)$ is the set of all distributions over $\mathcal{Y}$. In classification tasks, $\mathcal{Y}$ is discrete and $p_t$ is a multinomial distribution, and in regression tasks in which $\mathcal{Y}$ is a continuous domain, $p_t$ is usually a parametric probability density function, e.g. a Gaussian. For regression, we denote by $F_t : \mathcal{Y} \to [0, 1]$ the CDF corresponding to $p_t$.

According to (Kuleshov et al., 2018) a forecaster in a regression setting $H$ is calibrated if:

$$\frac{\sum_{t=1}^T \mathrm{I}\{y_t \leq F_t^{-1}(p)\}}{T} \xrightarrow{T \to \infty} p, \forall p \in [0, 1] \tag{1}$$

.

Intuitively this means that the $y_t$ is smaller than $F_t^{-1}(p)$ with probability approximately $p$, or that the predicted CDF matches the empirical one as the dataset size goes to infinity. This is equivalent to

$$P_{X,Y}\left(Y \leq [F(X)]^{-1}(p)\right) = p, \forall p \in [0,1] \tag{2}$$

Where $F(X)$ represents the CDF corresponding to H(X). This notion is translated by (Kuleshov et al., 2018) to a practical evaluation and calibration methodology. A re-calibration dataset $S = \{(x_t, y_t)\}_{t=1}^T$ is used to compute the empirical CDF value for each predicted CDF value $p \in F_t(y_t)$:

$$\hat{P}(p) = \frac{|\{y_t | F_t(y_t) \leq p, t = 1 \ldots T\}|}{T} \tag{3}$$

The calibration consists of fitting a regression function $R$ (i.e. isotonic regression) , to the set of points $\{(p, \hat{P}(p))\}_{t=1}^T$. For diagnosis the authors suggest a calibration plot of $\{(p, \hat{P}(p))\}$ at equally spaced values of $p$.

We start by intuitively explaining the basic limitation of this methodology. From Eq. 3 $\hat{P}$ is non-decreasing and therefore isotonic regression finds a perfect fit. Therefore, the modified CDF $R \circ F_t$ will satisfy $\hat{P}(p) = p$ on the re-calibration set, and the new forecaster is calibrated up to sampling error. This means that perfect calibration is possible no matter what the CDF output is, even for output CDFs which are statistically *independent* of the actual empirical uncertainty. We note that this might be acceptable when the uncertainty prediction is degenerate, e.g. all output distributions are Gaussian with the same variance, but this is not the case here. We also note that the issue is with the calibration definition not the re-calibration, as we show with the following analytic example.

We next present a concise analytic example in which the output distribution and the ground truth distribution are independent, yet fully calibrated according to Eq. 2. Consider the case where the target has a normal distribution $y_t \sim \mathcal{N}(0,1)$ and the network output $H(x_t)$ has a Cauchy distribution with zero location parameter and random scale parameter $\gamma_t$ independent of $x_t$ and $y_t$, defined as:

$$\begin{aligned} z_t &\sim & \mathcal{N}(0,1) \\ \gamma_t &= & |z_t| \\ H(x_t) &= & Cauchy(0, \gamma_t) \end{aligned} \tag{4}$$

Following a known equality for Cauchy distributions, the CDF output of the network $F_t(y) = F\left(\frac{y}{\gamma_t}\right)$, where $F$ is the CDF of a Cauchy distribution with zero location and 1 scale parameters. First we note that $\frac{y_t}{\gamma_t}$ and $\frac{y_t}{z_t}$, i.e. with and without the absolute value, have the same distribution due to symmetry. Next we recall the well known fact that the ratio of two independent normal random variables is distributed as Cauchy with zero location and 1 scale parameters (i.e. $\frac{y_t}{z_t} \sim Cauchy(0,1)$). This means that probability that $F_t(y_t) \equiv F(\frac{y_t}{\gamma_t}) \leq p$ is exactly $p$ (recall that $F$ is a $Cauchy(0,1)$ CDF). In other words, the prediction is perfectly calibrated according to the definition in Eq. 2, even though the scale parameter was random and independent of the distribution of $y_t$.

While the Cauchy distribution is a bit unusual due to the lack of mean and variance, the example does not depend on it and it was chosen for simplicity of exposition. It is possible to prove the existence of a distribution whose product of two independent samples is Gaussian (Thorin, 1977) and replace the Cauchy with a Gaussian, but it is an implicit construction and not a familiar distribution.

## 3  OUR METHOD

We present a new definition for calibration for regression, as well as several evaluation measures and a reliability diagram for calibration diagnosis, analogous to the ones used for classification (Guo et al., 2017). The basic idea is that for each value of uncertainty, measured through standard deviation $\sigma$, the expected mistake, measured in mean square error (MSE), matches the predicted error $\sigma^2$. This is similar to classification with MSE replacing the role of mis-classification error. More formally, if $\mu(x)$ and $\sigma(x)^2$ are the predicted mean and variance respectively then we consider a regressor well calibrated if:

$$\forall \sigma : \; \mathbb{E}_{x,y} \left[ (\mu(x) - y)^2 | \sigma(x)^2 = \sigma^2 \right] = \sigma^2. \tag{5}$$

In contrast to to (Kuleshov et al., 2018) this does not average over points with different values of $\sigma^2$ (at least in the definition, for practical measures some binning is needed), but only considers the mean and variance and not the entire distribution. We claim that this captures the desired meaning of calibration, i.e. for each *individual* example you can correctly predict the expected mistake.

Since we can expect each exact value of $\sigma^2$ in our dataset to appear exactly once, we evaluate eq. 3 empirically using binning, same as for classification. Formally, let $\sigma_t$ be the standard deviation of predicted output PDF $p_t$ and assume without loss of generality that the examples are ordered by increasing values of $\sigma_t$. We also assume for notation simplicity that the number of bins, $N$, divides the number of examples, $T$. We divide the **indices** of the examples to $N$ bins, $\{B_j\}_{j=1}^N$, such that: $B_j = \{(j-1) \cdot \frac{T}{N} + 1, \ldots, j \cdot \frac{T}{N}\}$. Each resulting bin therefore represents an interval in the standard deviation axis: $[\min_{t \in B_j}\{\sigma_t\}, \max_{t \in B_j}\{\sigma_t\}]$. The intervals are non-overlapping and their boundary values are increasing.

To evaluate how calibrated the forecaster is, we compare per bin $j$ two quantities as follows. The root of the *mean variance*:

$$mVAR(j) = \sqrt{\frac{1}{|B_j|} \sum_{t \in B_j} \sigma_t^2} \tag{6}$$

And the empirical *root mean square error*:

$$RMSE(j) = \sqrt{\frac{1}{|B_j|} \sum_{t \in B_j} (y_t - \hat{y}_t)^2} \tag{7}$$

where $\hat{y}_t$ is the mean of the predicted PDF ($p_t$)

For diagnosis, we propose a reliability diagram which plots the $RMSE$ as function of the $mVAR$ as shown in Figure 4. The idea is that for a calibrated forecaster per bin the $mVAR$ and the observed $RMSE$ should be approximately equal, and hence the plot should be close to the identity function. Apart from this diagnosis tool which as we will show is valuable for assessing calibration, we propose additional scores for evaluation.

**Expected Normalized Calibration Error (ENCE).** For summarizing the error in the calibration we propose the following measure:

$$ENCE = \frac{1}{N} \sum_{j=1}^N \frac{|mVAR(j) - RMSE(j)|}{mVAR(j)} \tag{8}$$

This score averages the calibration error in each bin, normalized by the bin's mean predicted variance, since for larger variance we expect naturally larger errors. This measure is analogous to the expected calibration error (ECE) used in classification.

**STDs Coefficient of variation ($C_V$).** In addition to the calibration error we would like to measure the dispersion of the predicted uncertainties. If for example the forecaster predicts a single homogeneous uncertainty measure for each example, which matches the empirical uncertainty of the predictor for the entire population, then the $ENCE$ would be zero, but the uncertainty estimation per example would be uninformative. Therefore, we complement the $ENCE$ measure with the Coefficient of Variation ($c_v$) for the predicted STDs which measures their dispersion:

$$c_v = \frac{\sqrt{\frac{\sum_{t=1}^T (\sigma_t - \mu_\sigma)^2}{T-1}}}{\mu_\sigma} \tag{9}$$

where $\mu_\sigma = \frac{1}{T} \sum_{t=1}^T \sigma_t$. Ideally the $c_v$ should be high indicating a disperse uncertainty estimation over the dataset. We propose using the $ENCE$ as the primary calibration measure and the $c_v$ as a secondary diagnostic tool.

### 3.1 CALIBRATION

To understand the need for calibration, let us start by considering a trained neural network for regression, which has very low mean squared error (MSE) on the train data. We now add a separate branch that predicts uncertainty as standard deviation, which together with the original network output interpreted as the mean, defines a Gaussian distribution per example. In this case, the NLL loss on the train data can be minimized by lowering the standard deviation of the predictions, without changing the MSE on train or test data. On test data however, MSE will be naturally higher. Since the predicted STDs remain low on test examples, this will result in higher NLL and ENCE values for the test data. This type of miss-calibration is defined as over-confidence, but opposite or mixed cases can occur depending on how the model is trained.

**Negative log-likelihood**. $NLL$ is a standard measure for a probabilistic model's quality (Hastie et al., 2001). When training the network to output classification confidence or a regression distribution, it is commonly used as the objective function to minimize. It is defined as:

$$NLL = -\sum_{t=1}^{T} log\left([H(x_t)](y_t)\right) \tag{10}$$

We propose using the NLL on the re-calibration set as our objective for calibration, and the reliability diagram, together with its summary measures ($ENCE$, $c_v$) for diagnosis of the calibration. In the most general setting a calibration function maps predicted PDFs to calibrated PDFs: $R(\Theta) : \mathcal{P}(\mathcal{Y}) \to \mathcal{P}(\mathcal{Y})$ where $\theta$ is the set of parameters defining the mapping.

Optimizing calibration over the re-calibration set is obtained by finding $\theta$ yielding minimal NLL:

$$\arg\min_{\theta}\left(-\sum_{t=1}^{T} log\left([R(p_t; \Theta)](y_t)\right)\right). \tag{11}$$

To ensure the calibration generalization, the diagnosis should be made on a separate validation set. Multiple choices exist for the family of functions $R$ belongs to. We propose using *STD Scaling*, (in analogy to Temperature Scaling (Guo et al., 2017)), which essentially multiplies the STD of each predicted distribution by a constant scaling factor $s$. If the predicted PDF is that of a Gaussian distribution, $\mathcal{N}(\mu, \sigma^2)$, then the re-calibrated PDF is $\mathcal{N}(\mu, (s \cdot \sigma)^2)$. Hence, in this case the calibration objective (Eq. 11) is:

$$\arg\min_{s}\left(-\sum_{t=1}^{T} log\left(\frac{1}{\sqrt{2\pi}s\sigma_t}e^{\frac{(y_t-\mu_t)^2}{2s^2\sigma_t^2}}\right)\right) = \arg\min_{s}\left(\frac{T}{2}log(s) - \sum_{t=1}^{T}\frac{(y_t-\mu_t)^2}{2s^2\sigma_t^2}\right) \tag{12}$$

If the original predictions are overconfident, as common in neural networks, then the calibration should set $s > 1$. This is analogous to Temperature Scaling in classification: a single multiplicative parameter is tuned to fix over or under-confidence of the model, and it does not modify the model's final prediction since $\mu_t$ remains unchanged.

**More complex calibration methods.** Histogram binning and Isotonic Regression applied to the STDs can be also used as calibration methods. We chose STD scaling since: (a) it is less prone to overfit the validation set, (b) it does not enforce minimal and maximal STD values, (c) it is easy to implement and (d) empirically, it produced good calibration results.

## 4 EXPERIMENTAL RESULTS

We next show empirical results of our approach on two tasks: a controlled synthetic regression problem and object detection bounding box regression. In both tasks we examine the effect of outputting trained and random uncertainty on the calibration process. In all training and optimization stages we use an SGD optimizer with learning rate $0.001$ and $0.9$ momentum.

### 4.1 SYNTHETIC REGRESSION PROBLEM

Experimenting with a synthetic regression problem enables us to control the target distribution $Y$ and to validate our method. We randomly generate $T = 50,000$ input samples $\{x_t, y_t\}_{t=1}^T$. We sample $x_t$

from $X \sim Uniform[0.1, 1]$ and $y_t$ from $Y \sim \mathcal{N}(x_t, x_t^2)$. This way, the target standard deviation of sample $x_t$ is $x_t$. We train a fully-connected network with four layers and a ReLU activation function on the generated training set using the smooth $\mathcal{L}_1$ loss function. We then add a separate branch with its own four layers to predict uncertainty.

Per example $x_t$, The original network output is considered the mean of a Gaussian distribution ($\mu_t$) and the additional output as its standard deviation ($\sigma_t$). For numerical stability, as suggested by (Kendall & Gal, 2017), the network outputs $log(\sigma^2)$. In the ***random uncertainty*** experiment, per example, the standard deviation representing the uncertainty is randomly drawn from $Uniform[1, 10]$. For the ***predicted uncertainty*** experiment, the uncertainty branch is optimized using the $NLL$ loss (Eq. 10) while the rest of the network weights are fixed. By fixing the remaining weights, the predicted mean ($\mu_t$) remains unchanged making sure we do not calibrate using over-confident predictions. We then re-calibrate as described in Sec. 3.1 on a separate re-calibration set consisting of $6,000$ samples.

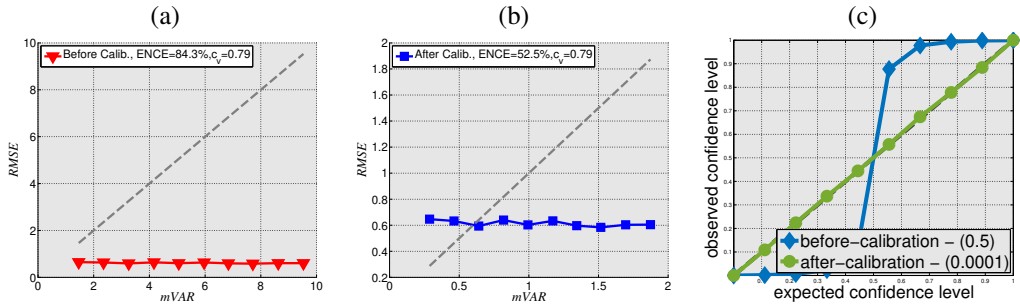

Figure 2: Reliability diagrams for the synthetic regression problem with **random uncertainty estimation**. Reliability diagram using our method before (a) and after (b) calibration. (c) Before and after calibration based on the confidence intervals method (Kuleshov et al., 2018).

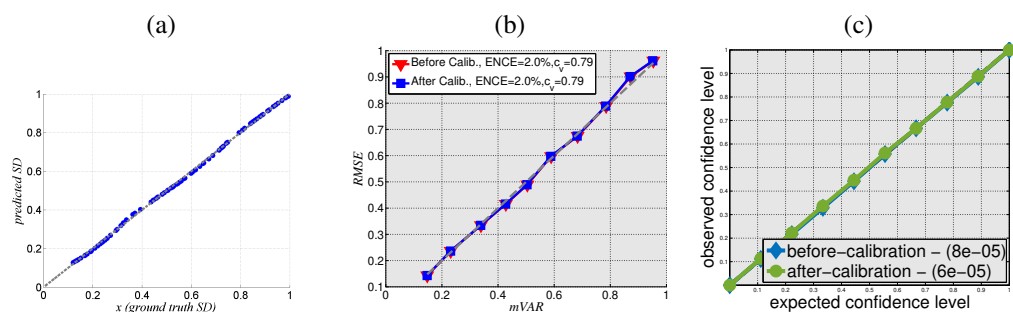

Figure 3: Synthetic regression problem with **predicted uncertainty**. (a) ground truth vs. predicted standard deviation. (b) Reliability diagram before and after calibration. (c) Reliability diagram using confidence intervals (Kuleshov et al., 2018) before and after calibration.

As one can see in Fig. 2 the confidence interval method can almost perfectly calibrate the ***random*** independent uncertainty estimation, as the expected and observed confidence level match and we get the desired identity curve. This phenomenon is extremely undesirable for safety critical applications where falsely relying on uninformative uncertainty can lead to severe consequences. It is important to note that the perfect calibration did not arise from giving the same fixed $\sigma$ for each prediction, which would be acceptable, as the isotonic regression modifies the probabilities directly and not the outputted standard deviations. In contrast you can see how our method can only marginally improve the calibration and one can clearly see, both from the ENCE value and visually from the graph, that the predictions are not calibrated. In the trained experiment, in which uncertainty is ***predicted*** by the network, we can see in Fig. 3 that the network almost perfectly learns the correct uncertainty, as expected from the problem simplicity and the high data availability. In this case both methods do not change the calibration results much. The important thing to note is that our calibration and evaluation

method can easily differentiate between both cases, the random and predicted uncertainty, while they are almost exactly the same after calibrating with (Kuleshov et al., 2018).

## 4.2 BOUNDING BOX REGRESSION FOR OBJECT DETECTION

In computer vision, an object detector outputs per input image a set of bounding boxes, each commonly defined by 5 outputs: classification confidence and four positional outputs $(t_x, t_y, t_w, t_h)$ representing its (x,y) position, width and height. We show results on each positional output as an independent regression task using the R-FCN detector (Dai et al., 2016). To this architecture we add an additional uncertainty branch predicting the corresponding STDs for each regression output, $(\sigma_x, \sigma_y, \sigma_w, \sigma_h)$. Thus, the network outputs a Gaussian distribution per regression task. For training the network weights we use the entire Common objects in context (COCO) dataset (Lin et al., 2014). For uncertainty calibration and validation we use two separate subsets of the KITTI (Geiger et al., 2012) object detection benchmark dataset, which consists of road scenes. Training the uncertainty output on one dataset and performing calibration on a different one reduces the risk of over-fitting and increases the calibration validity. See Appendix A for further details.

We initially train the network without the additional uncertainty branch as in (Dai et al., 2016), while the uncertainty branch weights are randomly initialized. Therefore, in this state which we refer to as ***untrained uncertainty***, random uncertainties are assigned to each example. We then train the uncertainty branch by minimizing the $NLL$ loss (Eq. 10) on the training set, freezing all network weights but the uncertainty head for $1K$ training iterations with 6 images per iteration. Freezing the rest of the network ensures that the additional uncertainty estimation does not sacrify accuracy. The result of this stage is the network with ***predicted uncertainty***. Finally, we train the $NLL$ loss for $1K$ additional training iterations on the *re-calibration set*, to optimize the single scaling parameter $s$, and obtain the ***calibrated uncertainty***.

Figure 4 shows the resulting reliability diagrams before calibration (***predicted uncertainty***) and after (***calibrated uncertainty***) for all four positional outputs, on the validation set. As can be observed from the monotonously increasing curve before calibration, the output uncertainties are indeed correlated with the empirical ones. Additionally, since the curves are entirely above the ideal one, the predictions are over confident. Using the learned scaling factor $s$ which varies between 1.1 and 1.2, the $ENCE$ is significantly reduced as shown in table 1. The $c_v$ remains unchanged after calibration since it is invariant to uniform scaling of the output STDs (Eq. 9). For ***untrained uncertainty***, Fig. 1 shows that after calibration, just as with the synthetic dataset, using the *interval-based* method, uncertainty is almost perfectly calibrated. In contrast, our method reveals the lack of correlation between the predictions and empirical uncertainties before/ after applying calibration (See results in Appendix A).

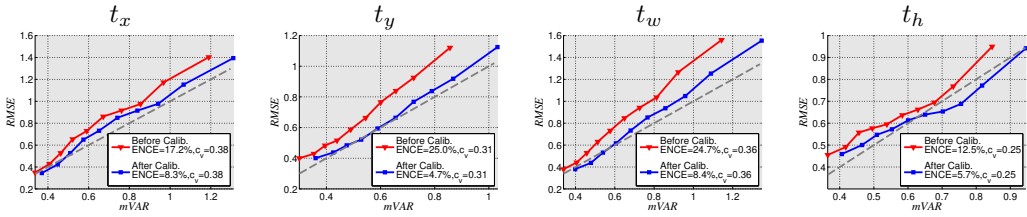

Figure 4: **Reliability diagrams for bounding box regression on the KITTI validation set before and after calibration.** Each plot compares the empirical $RMSE$ and the root mean variance ($mVAR$) in each bin. Grey dashed line indicates the ideal calibration line. See Sec. 4.2 for details.

## 5 CONCLUSIONS

Calibration, and more generally uncertainty prediction, are critical parts of machine learning especially in safety-critial applications. In this work we exposed serious flaws in the current approach to define and evaluate calibration for regression problem. We also proposed an alternative approach and showed that even a very simple re-calibration method can lead to significant improvement in real-world applications. Our proposed method for calibration effectively takes into consideration the first two moments when comparing the output and real distributions. An interesting direction for future

Table 1: Evaluation of uncertainty calibration for the bounding box regression tasks on the KITTI validation dataset.

|  | Before calibration | | After calibration | |
| --- | --- | --- | --- | --- |
|  | $ENCE$ | $C_v$ | $ENCE$ | $C_v$ |
| $\mathbf{t_x}$ | 17.2% | 0.38 | 8.3% | 0.38 |
| $\mathbf{t_y}$ | 25.0% | 0.31 | 4.7% | 0.31 |
| $\mathbf{t_w}$ | 24.7% | 0.36 | 8.4% | 0.36 |
| $\mathbf{t_h}$ | 12.5% | 0.25 | 5.7% | 0.25 |

research would be extending our method to handle more complex distributions using, for example, higher-order moments.

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

# A  BOUNDING BOX REGRESSION FOR OBJECT DETECTION: ADDITIONAL DETAILS AND RESULTS

In this section we provide additional details on the network architecture and datasets used for bounding box prediction as well as additional results using random predictions. As our base architecture we use the R-FCN detector (Dai et al., 2016) with a ResNet-101 backbone (He et al., 2016). The R-FCN regression branch outputs per region candidate a 4-d vector that parametrizes the bounding box as $t_b = (t_x, t_y, t_w, t_h)$ following the accepted parametrization in (Girshick, 2015). We use these outputs in our experiments as four seperate regression outputs. To this architecture we add an *uncertainty branch*, *identical in structure to the regression branch*, that outputs a 4-d vector $(u_1, u_2, u_3, u_4) \equiv (log(\sigma_x^2).log(\sigma_y^2), log(\sigma_w^2), log(\sigma_h^2))$, each representing the log variance of the Gaussian distributions of the corresponding output. As before, the original regression output represents the Gaussian mean (i.e. $\mu_x = t_x$).

For training the network weights we use the entire Common objects in context (COCO) dataset (Lin et al., 2014). As stated previously we use a two-stage training approach. We first train the original R-FCN network, and then freeze all weights and train only the additional uncertainty prediction branch. In this way we train uncertainty prediction without sacrificing the network's accuracy. Note however that our method completely holds if the entire network is trained at once (e.g. if confidence estimation importance is such that accuracy may be marginally sacrified). For uncertainty calibration we use the KITTI (Geiger et al., 2012) object detection benchmark dataset, which consists of road scenes. We divide the KITTI dataset into a re-calibration set used for training the calibration parameters ($\sim 6K$ images), and a validation set ($\sim 1.5K$ images, $37K$ object instances). The classes in the KITTI dataset represent a small subset of the classes in the COCO dataset, and therefore we reduce our model training on COCO to the 9 relevant classes (e.g. car, person) and map them accordingly to the KITTI classes.

Figure 5 shows the reliability diagrams for the four bounding box regression outputs with ***untrained uncertainty*** before and after we apply our calibration method. As with the synthetic dataset, the graphs immediately reveal the disconnect between the random values and the empirical uncertainties. In all the cases the calibration results in a highly non-calibrated uncertainty according to our metrics.

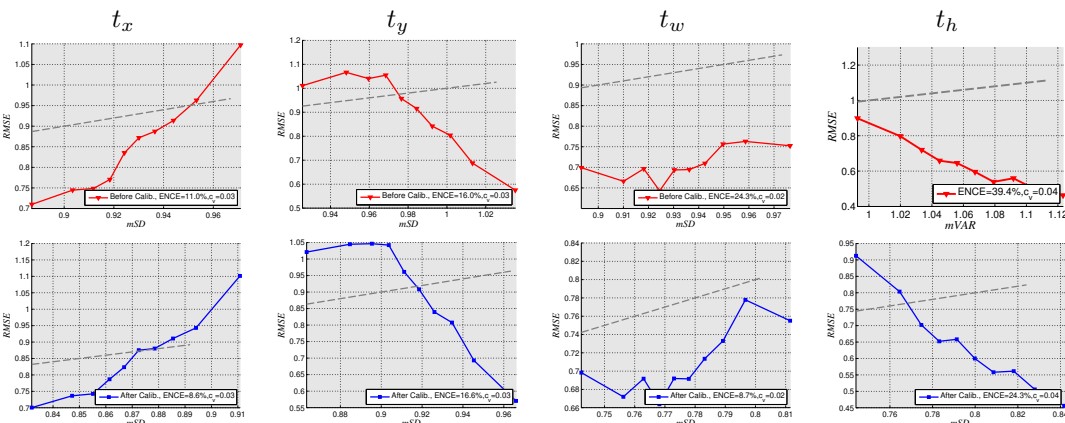

Figure 5: **Reliability diagrams for bounding box regression with untrained uncertainty estimation for the bounding box regression outputs** $(t_x, t_y, t_w, t_h)$**.** Top row: before calibration, bottom row: after calibration.

