# OpenReview forum: "Evaluating and Calibrating Uncertainty Prediction in Regression Tasks"
_ICLR.cc/2020/Conference — Reject_

### Official Review · AnonReviewer1 · 2019-10-20
**Official Blind Review #1**

**Rating:** 1

**Review:**

First, I do not follow the authors’ argumentation in the motivating example where the discus calibration in a mis-specified model. They argue that it should not be possible to calibrate the output of a mis-specified model. It is of course a bad idea to model a Gaussian with a Cauchy distribution (the way it is always a bad idea to have a mis-specified model), but I would actually argue that the isotonic regression based approach by Kuleshov et al., 2018 is more robust against model mis-specification due to its higher flexibility and thus a more powerful approach for pos-hoc calibration.

Second, I find the approach lacks novelty and is a straightforward application of the well established temperature scaling method and the expected calibration error to regression tasks. In addition, there are only minor differences to Phan et al 2018, CoRR.

Most importantly, I have major concerns regarding the lack of in-depth evaluation. The authors provide no systematic comparison to the state-of-the art (Kuleshov et al., 2018) and only show reliability diagrams (they introduced themselves) for one singe (real) dataset and only their own method.

**Experience Assessment:**

I have published one or two papers in this area.

**Review Assessment: Checking Correctness Of Derivations And Theory:**

I carefully checked the derivations and theory.

**Review Assessment: Checking Correctness Of Experiments:**

I carefully checked the experiments.

**Review Assessment: Thoroughness In Paper Reading:**

I read the paper thoroughly.

---

### Official Review · AnonReviewer3 · 2019-10-20
**Official Blind Review #3**

**Rating:** 3

**Review:**

This paper focuses on the calibration for the regression problem. First, it investigates the shortcomings of a recently proposed calibration metric [1] for regression, and show theoretically where this metric can be fooled. Later, it introduces new metrics for measuring the calibration in regression problem which are instantiated from similar idea as the ECE calibration metric [2] used for classification problem. The paper defines the uncertainty as the mean square error and like the ECE idea, it divides the samples into different uncertainty bins and for each bin, it calculates RMSE of network output uncertainty. The RMSE versus variance of estimated uncertainty is depicted as the reliability diagram.
To summarize the calibration error as the numbers, the paper proposes two other metrics as ENCE and CV that explain the overall calibration error for the network and calibration error for each sample, respectively. It also shows the effectiveness of the metrics, on the synthetic data where the calibration metric proposed in [1] can be fooled VS. more informative metrics like proposed reliability diagram with RMSE and mVAR.  The other experiment is conducted on object detection problem where the uncertainty of bounding box positional outputs is considered as the calibration for the regression problem.


Pros:
1- Investigating theoretically the shortcoming of the calibration metric for regression is interesting
2- Proposing the new metrics specifically  designed for regression in novel, however it needs more accurate investigation.

Cons:
The paper is not well-written and the experiments and discussions are not support the ideas, more specifically I can mention several concerns:

1- Motivation: The flaw of the previously proposed calibration metric is not explained clearly. The paper only discusses in which scenario, the metric cannot work properly considering. However, the considered assumption (for instance the output of uncertainty estimation from the network has uniform distribution) can never happen in real scenario. Then the importance of redefining of calibration method is not clear.

2-  Accuracy of Proposed Method: The measure proposed for calibration (Eq. 8) should be shown it will converge to the definition of calibration proposed in Eq.5, which is missing in the paper.

3- Lack of Clarity in some Parts: The paper introduces the  new architecture for adding the uncertainty output to the regression network. But this new architecture is not explained clearly. I suggest the authors add more details about this part.

4- Justification: The paper trains the uncertainty part of the network with NLL loss function and later fine-tunes the output with temperature scaling method. But this calibration  method is not related to the proposed metric which is claimed it has better calibration clarity. Then the importance of using the new metric to define the calibration is not clear.

5- Experiments: The experiment setup needs more accuracy. The paper should investigate the shortcomings of the previous metric in the same setting as the new proposed metric. In experiment Sec. 4.1, the settings for obtaining the output of the network uncertainty is different then comparing the results are not fare.  The experiments for real scenario is not wide enough. It just shows that the parameters get calibrated. However we expect to see more results about the CV metric and its importance to define.

Overall, I think this paper should be rejected as 1) the novelty of proposed metrics are not enough. 2) the motivation and justification of why the proposed metrics is better than previous metrics is not clear. 3) the experiments are not supporting the idea.

References:

[1]  Kuleshov, Volodymyr, Nathan Fenner, and Stefano Ermon. "Accurate Uncertainties for Deep Learning Using Calibrated Regression." International Conference on Machine Learning. 2018.

[2] Naeini, Mahdi Pakdaman, Gregory Cooper, and Milos Hauskrecht. "Obtaining well calibrated probabilities using bayesian binning." Twenty-Ninth AAAI Conference on Artificial Intelligence. 2015.


**Experience Assessment:**

I have published one or two papers in this area.

**Review Assessment: Checking Correctness Of Derivations And Theory:**

I assessed the sensibility of the derivations and theory.

**Review Assessment: Checking Correctness Of Experiments:**

I carefully checked the experiments.

**Review Assessment: Thoroughness In Paper Reading:**

I read the paper at least twice and used my best judgement in assessing the paper.

---

### Official Review · AnonReviewer2 · 2019-10-25
**Official Blind Review #2**

**Rating:** 1

**Review:**

This paper is concerned with uncertainty calibration diagnostics and re-calibration methods, applied to neural network regression. It is motivated by a flaw in the diagnostic proposed by Kuleshov+ 2018 (abreviated K2018 below), as explained around eq4, and proposes a replacement diagnostic for uncertainty calibration quality (sec 3 before 3.1). It then specialises by considering the class of uncertainty prediction schemes qualified as "direct" uncertainty modeling (defined sec1), in which the network predicts the parameters of a parametric distribution over the target output, typically mean and variance of a Gaussian. For this class of schemes, it proposes a recalibration method (sec 3.1), which consists, as shown eq12, of rescaling (with a single parameter $s$) the square root of the variance predicted by the neural network. It then presents experiments (sec4) to demonstrate that the motivating flaw can be evidenced by their diagnostic, and fixed by their recalibration method where it makes sense (ie where predicted uncertainties are not random, i.e. statistically independent of the empirical uncertainty).

The formal treatment of the issue appears sound, as do the experiments. Nevertheless I believe the paper should be rejected for the following reasons. (1) I fail to see the relevance of the alleged flaw, as explained below. I may have missed something and am looking forward to the author's response. (2) The proposed palliatives are not easy to extend to other forms of uncertainty prediction, even for "direct" (parametric) uncertainty modelling schemes, as they are specialised (def of ENCE eq8 and STD coef of variation eq9). (3) The advantage, mentioned just below eq2, of not averaging over data points, breaks down immediately since binning is needed; this is a consequence of outputting probability distributions over continuous variables, in the case of K2018, and of outputting a continuous parameter, $\sigma^2$, in the present paper.

Regarding (1): I am not surprised that, using K2018's setup, it is possible to recalibrate (i.e. minimise a calibration measure), on a given calibration data set, a model that outputs random uncertainty distributions. The calibration metric alone is not sufficient to measure the goodness of the model: that should take into account prediction sharpness, or test NLL. For a model with random uncertainty output, recalibrated with K2018, I would expect that the miscalibration appears on these other metrics evaluated on the test set.

Suggestions for improvement (did not affect my assessment):
- the training procedure is mostly common to sec4.1 and sec4.2, and explained relatively clearly: on training set, first train mean using eg L2 loss, then variance output against NLL loss, then recalibration parameter against NLL loss. This procedure could be discussed more explicitly and jointly for both experiments.
- polishing the text and grammar

**Experience Assessment:**

I have read many papers in this area.

**Review Assessment: Checking Correctness Of Derivations And Theory:**

I assessed the sensibility of the derivations and theory.

**Review Assessment: Checking Correctness Of Experiments:**

I assessed the sensibility of the experiments.

**Review Assessment: Thoroughness In Paper Reading:**

I read the paper at least twice and used my best judgement in assessing the paper.

---

### Decision · Program_Chairs · 2019-12-19

**Decision:**

Reject

**Comment:**

The paper investigates calibration for regression problems. The paper identifies a shortcoming of previous work by Kuleshov et al. 2018 and proposes an alternative.

All the reviewers agreed that while this is an interesting direction, the paper requires more work before it can be accepted. In particular, the reviewers raised several concerns about motivation, clarity of the presentation and lack of in-depth empirical evaluation.

I encourage the authors to revise the draft based on the reviewers’ feedback and resubmit to a different venue.